

# Turbulent Energy Budget Analysis Based on Coherent Wind Lidar Observations

Jinhong Xian[1,2], Zongxu Qiu[2], Hongyan Luo[2], Yuanyuan Hu[2], Xiaoling Lin[2], Chao Lu[2], Yan Yang[2], Honglong Yang[2], Ning Zhang[1,3]

[1] School of Atmospheric Sciences, Nanjing University, Nanjing 210023, China
[2] Shenzhen National Climate Observatory, Meteorological Bureau of Shenzhen Municipality, Shenzhen 518040, China
[3] Key Laboratory of Urban Meteorology, China Meteorological Administration, Beijing, 100089, China

*Correspondence to*: Honglong Yang (yanghonglong@weather.sz.gov.cn); Ning Zhang (ningzhang@nju.edu.cn)

**Abstract.** The turbulent kinetic energy (TKE) budget term, as a key physical quantity for describing the generation and dissipation processes of turbulence, is crucial for revealing the essence and characteristics of turbulence. Due to limitations in current observational methods, the generation and dissipation mechanisms of atmospheric turbulent energy are mainly based on ground or tower-based observations, and studies on the budget terms of TKE of vertical structures are lacking. We propose a new method for detecting TKE budget terms based on coherent wind lidar, and compare it with data obtained with a three-dimensional ultrasonic anemometer. The results show that the lidar errors are relatively small, less than 0.0001 $m^2/s^3$, for at least 47% of the data, which verifies the accuracy and reliability of our method. We explore the generation and dissipation mechanisms of turbulence under different weather conditions, and find that the buoyancy generation term plays a role in dissipating TKE under low cloud and light rain conditions. During the day, turbulent transport and the dissipation rate are the main dissipation terms, while buoyancy generation is the main dissipation term at night. The results show that the proposed method can accurately capture the vertical distribution of TKE, dissipation rate, shear generation, turbulent transport, and buoyancy generation terms in the boundary layer, and can comprehensively reflect the influence of each budget term on the vertical structure of turbulent energy. This research provides a new perspective and method for studies of atmospheric turbulence, which can be further applied to fine observations of the vertical structure and dynamics of turbulence.

## 1 Introduction

Turbulence is an important phenomenon in atmospheric thermodynamics and dynamics. Its generation and dissipation mechanisms are of great significance for understanding atmospheric motion, predicting weather changes, and evaluating wind energy resources (Heilman et al., 2018; Stull, 1988; Byzova et al., 1989; Kaimal et al., 1976). The turbulent kinetic energy (TKE) budget term, as a key physical quantity for describing the generation and dissipation processes of turbulence, is crucial for revealing the essence and characteristics of turbulence (Stull, 1988).

Boundary layer parameterization schemes can be used to predict the vertical distribution of TKE and its budget terms by simulating turbulent processes, thereby allowing the impact of different physical processes on TKE changes to be analyzed (Nilsson et al., 2016b; Elguernaoui et al., 2023). The



boundary layer parameterization schemes currently used (YSU, MYJ, MYNN2, ACM2, etc.) make certain assumptions and simplifications when simulating turbulent processes, which inevitably lead to

inaccurate predictions of TKE (Hariprasad et al., 2014). For example, the YSU scheme tends to overestimate or underestimate mechanical turbulence, turbulent mixing intensity, or entrainment processes, thereby affecting the prediction results of TKE (Hong et al., 2006). The MYJ scheme underestimates the actual turbulence intensity when the TKE is predicted during the day, causing the predicted values to be smaller than the actual observed values, especially under strong convection or

complex terrain conditions (Janjic, 1994). For the MYNN2 parameterization scheme, the prediction of TKE under complex weather conditions is not accurate enough. Therefore, some authors sought to enhance these parameterization schemes through optimization (Hariprasad et al., 2014; Xie et al., 2013). Nilsson et al. proposed a simple model to describe the evolution of TKE and its budget under shear convective atmospheric conditions (Nilsson et al., 2016b). The core of this model is that the variation of

TKE is determined by four budget terms (turbulence dissipation rate, buoyancy generation, shear generation, and vertical transmission of TKE), and only three measurable input parameters (near surface buoyancy flux, boundary layer depth, and the wind speed at a certain height of the surface layer) are needed to predict the vertical distribution of TKE and its budget terms. However, in the atmospheric boundary layer, the generation and dissipation of turbulent energy are complex physical processes that

are influenced by various factors, including changes in surface heat flux, atmospheric stability, and topography. Especially for turbulence in the free convection boundary layer, its dynamic characteristics are quite complex, and there are significant differences between the results of models and measurement. Indeed, high-resolution observations are needed to reveal its inherent mechanisms (Goger et al., 2018; Lobocki, 2017).

60        In previous studies, observations of the energy generation and dissipation mechanisms of atmospheric turbulence were mainly based on near-ground or tower-based observations (Baas et al., 2018; Babic and Rotach, 2018; Barman et al., 2019; Bodini et al., 2018; Caughey and Wyngaard, 1979; Hang et al., 2020; Jensen et al., 2017; Lobocki, 2017; Tong et al., 2022; Yus-Diez et al., 2019). Nilsson et al. used near surface measurement data from a small tower to conduct a detailed analysis of various terms

in the TKE budget, including the trend, buoyancy generation, dissipation, and transport terms, revealing various differences in the surface layer dynamics of TKE and its attenuation at different periods of the day (Nilsson et al., 2016a). Canut et al. successfully measured turbulence flux and variance in the atmospheric boundary layer using an acoustic anemometer bound to a tethered balloon, and conducted an in-depth analysis of the variation patterns of these parameters (Canut et al., 2016). Li et al. analyzed

changes in the boundary layer structure, turbulence intensity, and flux of the sea breeze front (SBF) using observational data obtained from two meteorological towers (Tianjin Meteorological Tower and Beijing Meteorological Tower) (Li et al., 2023). Their study indicated that the passage of the SBF leads to an increase in mechanical turbulence, manifested by an increase in friction velocity and TKE, and the shear generation term in the TKE budget equation underwent a more significant increase than the buoyancy

generation term. Pozzobon et al. collected turbulence data at four different height levels (3, 6, 14, and 30 m) over a period of 10 months at a 30 m tower, and studied the TKE budget under convective daytime conditions and stable nighttime conditions (Pozzobon et al., 2023). Their results indicated that during the



day, the TKE budget was mainly dominated by shear and buoyancy generation, while dissipation was the main dissipation mechanism. At night, the TKE budget was mainly generated by shear, while buoyancy generation term played a dissipation role.

However, tower-based observations have certain drawbacks, such as a limited detection range, and previous research mainly focused on the near-ground TKE budget and its variational patterns. As such, there is still a lack of research on the various budget terms of TKE for vertical structures. With the rapid development of meteorological observation technology, laser wind measurement technology has gradually become an important tool in the field of wind field observations due to its high precision, high resolution, and wide detection range. Especially in the study of atmospheric turbulence, the application of coherent wind lidar has demonstrated its unique advantages (Banakh et al., 2021; Banakh, 2013; Xian et al., 2023; Rios and Ramamurthy, 2023). In our previous studies, the direct acquisition of atmospheric turbulence parameters was achieved based on wind lidar (Xian et al., 2024b; Xian et al., 2024a). On this basis, we developed a new detection method for TKE budget terms to study the generation and dissipation mechanism of turbulent energy in the vertical direction of the boundary layer, the results of which are presented in this study.

The remainder of this paper is organized as follows. In Section 2, we introduce the instrument and data quality control methods. In Section 3, we introduce methods for obtaining various TKE budget terms based on wind lidar data and compare them with the data obtained with a three-dimensional ultrasonic anemometer to verify the accuracy of the proposed method. In Section 4, we use the measured data to analyze the spatiotemporal variation characteristics of the TKE budget term in classical cases, and explore the variational patterns of each budget term under different weather conditions. The main conclusions of this study are presented in Section 5.

## 2 Instruments and Data Quality Control

The Shiyan Observation Base (113.90586 °E, 22.65562 °N) is located in the remote suburbs of Shenzhen, with no large obstacles (such as high-rise buildings) around it. One to two km northeast of the base is farmland, while the terrain to the south and northwest is generally flat, almost completely covered by forests and lakes, as shown in Figure 1(a). The base has the highest meteorological gradient observation tower in Asia, which is 356 m high. Due to its advantageous geographical location and the absence of obstacles around it, monitoring data obtained with the three-dimensional ultrasonic anemometer and thermometer on the gradient observation tower have strong representativeness (Zhou et al., 2023). Ultrasonic anemometers (CSAT3, Campbell Scientific, Utah, USA) capable of simultaneously observing the wind speed and temperature in three dimensions were installed on the tower at heights of 160 m and 320 m. The observation frequency of the ultrasonic anemometer is 10 Hz, with a wind speed accuracy of 0.1 m/s and a temperature accuracy of 0.002 K. A coherent wind lidar was installed under the gradient observation tower, as shown in Figure 1(b). The wind lidar (DSL-W, Darsunlaser Technology Co., Ltd., Shenzhen, China) has a detection blind zone of only 30 m, with a maximum detection altitude of 3 km and a vertical resolution of 30 m. The time resolution is 5 s, which means the observation frequency is 0.2 Hz. Specific performance indicators of the laser anemometer and the three-dimensional ultrasonic anemometer are shown in Table 1.



The following quality control steps were taken for the wind speed and temperature data obtained with the three-dimensional ultrasonic anemometer.

(1)     *Calculation of statistical parameters*. Calculate the average and standard deviation of the observed values using 30 min intervals.

(2)     *Outlier identification and handling*. Any observed value that deviates more than three times the standard deviation from the mean is marked as outlier data and designated as a missing value. This is based on the commonly used "triple standard deviation principle" in statistics, which means that the probability of data points falling outside the mean plus or minus three standard deviations is very small (close to zero), and therefore, these points are considered outliers. This process is repeated three times to ensure accurate identification and handling of outliers.

(3)     *Data loss rate*. Within 30 min, if the number of lost measurements exceeds 20%, the data for that period are discarded.

(4)     *Coordinate axis correction*. To eliminate wind speed errors caused by installation tilt errors of the ultrasonic anemometers, a dual rotation method is used to correct the coordinate axis, ensuring the accuracy and reliability of the wind speed data (Zhou et al., 2023).

Next, we undertook the following quality control steps for the wind lidar data.

(1)     *Regular inspection*. The wind speed measurements are checked every 30 min every day.

(2)     *Outlier removal*. We identify outliers and eliminate data points that deviate from the mean by more than three standard deviations, following the same method as described above.

(3)     *Integrity of the profile data*. For a single wind speed profile data set, if more than 20% of data points below 500 m are lost, the entire profile is discarded to ensure data integrity and reliability.

*Turbulence stationarity test*. When studying the characteristics of atmospheric boundary layer turbulence, methods such as correlation analysis and spectral analysis are usually employed. These analysis methods are usually based on the premise that atmospheric turbulence fluctuations have relatively stable statistical characteristics over a certain period, which is called "stationarity" [29]. The requirement for stationarity is that the main statistical characteristics of turbulence, such as variance, should remain stable during the selected observation period. In short, this means that throughout the entire observation period, the average of the overall variance should be close to the average of the variances in each shorter period (Massman, 2006). In this study, a strategy is adopted to compare the average variance within a 30 min observation period with the average variance of six 5 min samples within the same period. By calculating the deviation between these two and setting a threshold (such as less than 0.3), we filter the data to ensure the reliability of the subsequent analysis.

Overall, these data quality control processes aim to ensure the accuracy, completeness, and reliability of the data through statistical methods and physical corrections. This is crucial for accurate inference of the turbulence parameters, as their values largely depend on the accuracy of the wind speed measurements and the assumption of stationarity.



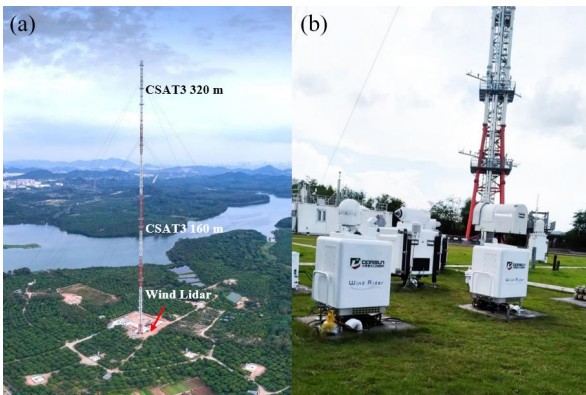

**Figure 1.** The three-dimensional ultrasonic wind and temperature sensors installed at the meteorological gradient observation tower (a) and the wind lidar used to make wind measurements, located under the tower (b).

**Table 1.** Performance parameters of the ultrasonic anemometer and wind lidar instrument

|  | Metric | Technical Performance Requirements |
| --- | --- | --- |
| Ultrasonic anemometer | Observational frequency | 10 Hz |
|  | Resolution of the wind speed | ≤0.1 m/s |
|  | Resolution of the wind direction | ≤1° |
|  | Range of wind speed measurements | 0–40 m/s |
|  | Temperature accuracy | ≤0.002 K |
| Wind Lidar | Minimum detection altitude | ≤30 m |
|  | Maximum detection altitude | 3 km |
|  | Distance resolution | 30 m |
|  | Observational frequency of the wind profile | 0.2 Hz |
|  | Resolution of the wind speed | ≤0.1 m/s |
|  | Resolution of the wind direction | ≤1° |
|  | Range of wind speed measurements | 0–60 m/s |
|  | Range of wind direction measurements | 0°– 360° |

160

To verify the accuracy of the data that passed the above quality control processes, a comparison was made between the three wind speed components ($u$, $v$, $w$) measured by the ultrasonic anemometer and the wind lidar, as shown in Figure 2. It can be seen from the figure that the two data sets have relatively high consistency, which lays a solid foundation for the subsequent analysis.



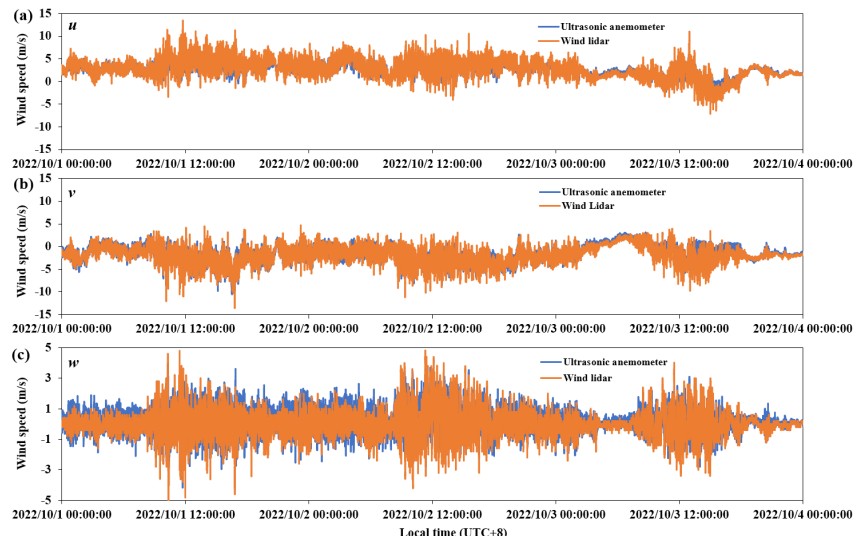

**Figure 2.** Comparison of the three wind speed components measured by the ultrasonic anemometer (blue) and wind lidar (orange).

## 3 Methodology

### 3.1 Theory

The TKE equation can be expressed as (Stull, 1988; Nilsson et al., 2016a)

$$
\underbrace{\frac{\partial E}{\partial t}}_{E_t} = \underbrace{-\overline{u'w'}\frac{\partial u}{\partial z} - \overline{v'w'}\frac{\partial v}{\partial z}}_{S} + \underbrace{\frac{g}{\theta_v}\overline{w'\theta'}}_{B} - \underbrace{\frac{\partial \overline{w'E'}}{\partial z}}_{T_t} - \underbrace{\frac{\partial \overline{w'p'/\rho_0}}{\partial z}}_{T_p} - \underbrace{\varepsilon}_{D}, \tag{1}
$$

where $E$ represents TKE (m²/s²), $t$ is the time (s), and $u'$, $v'$, and $w'$ are the fluctuation values of the three-dimensional wind speed components $u$, $v$, and $w$, respectively, which vary with height $z$; $g$ is gravitational acceleration, $\theta_v$ is the average absolute temperature, $\theta'$ is the fluctuation value of the absolute temperature $\theta$, $\rho_0$ is the air density, $P'$ is the fluctuation value of the air pressure $P$, and $\varepsilon$ is the average dissipation rate of TKE. On the left side of the equation is the tenancy term ($Et$), while on the right side are the budget terms for shear generation ($S$), dissipation rate ($D$), turbulent transport ($T_t$), pressure transport ($T_P$), and buoyancy generation ($B$). Next, we will introduce how to obtain the above budget terms based on wind lidar data.

### 3.2 Determination of Turbulent Kinetic Energy and the Tenancy Term

TKE can also be expressed as

$$
E = \frac{1}{2}(u'^2 + v'^2 + w'^2). \tag{2}
$$

These fluctuation components can be obtained by subtracting the average of the observed wind speed data within a time window of duration $N$. In this study, a phase window of $N \approx 20$ min is used.



Figure 3(a) shows a comparison of TKE obtained with wind lidar at heights of 150 m and 160 m (both shown in orange) and with the three-dimensional ultrasonic anemometer (shown in blue) from October 1 to 9, 2022. From the figure, it can be seen that the two data sets have a very high degree of consistency. By taking the time derivative of TKE($\Delta t \approx 5$ s), a comparison of the tenancy term obtained with both sets of instruments is obtained, as shown in Figure 3(b). It can be seen from this plot that the wind lidar data can reflect the trend of turbulence changes very well.

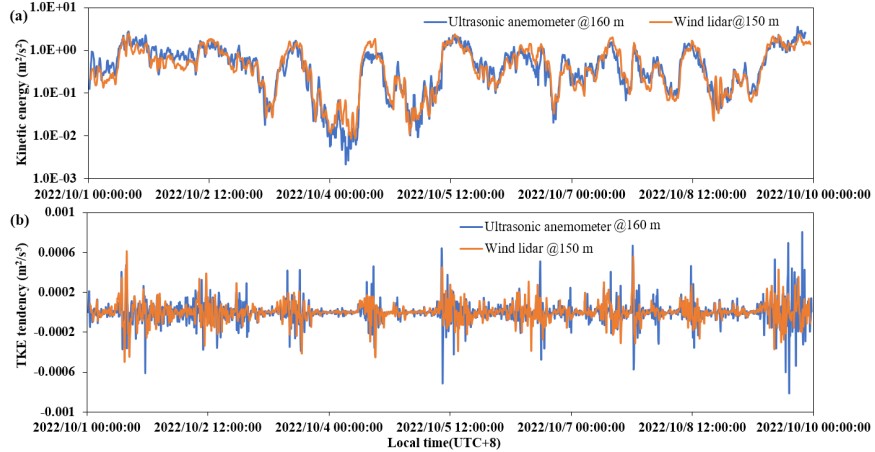

**Figure 3.** Comparison of TKE (a) and tenancy term (b) obtained with the ultrasonic anemometer (blue) and wind lidar (orange).

### 3.3 Determination of the Dissipation Rate

Calculation of the dissipation rate is a major challenge. In our previous work, we proposed a method based on wind lidar to detect atmospheric turbulent energy directly (Xian et al., 2024b; Xian et al., 2024a). In the current study, we directly obtain the dissipation rate based on this foundation. According to Kolmogorov's theory of a local homogeneous and isotropic medium (Kolmogorov, 1941), the turbulence spectrum can be represented as

$$P(k) = c\varepsilon^{2/3} k^n , \tag{2}$$

$$k = \frac{2\pi f}{U} , \tag{3}$$

where the units of $P$ are m³/s², $c$ is the Kolmogorov constant (0.55), $\varepsilon$ is the turbulent kinetic dissipation rate (m²/s³), $n$ is the power-law exponent, $f$ is the frequency (1/s), $k$ is the wave number (1/m), and $U$ is the average wind speed (m/s). The turbulence spectrum $P$ is obtained by performing fast Fourier transform on data with a time window of 20 min. In the case of a known spectrum, the dissipation rate can be obtained by the following formula

$$\varepsilon = \{\frac{P(k_1) - P(k_0)}{c(k_1^n - k_0^n)}\}^{3/2} . \tag{4}$$





In our previous work, we demonstrated that wind lidar can reflect the spectral characteristics of
TKE in the range of $f$ from $10^{-2.5}$ to $10^{-1}$ Hz; therefore, in this study these two values are used for $k_0$ and
$k_1$, respectively (Xian et al., 2024b). According to Kolmogorov's theory of local isotropic turbulence, the
value of $n$ is –5/3, which is also known as the "–5/3 power law." In a previous study, we revealed that
the distribution of the power-law exponent in the vertical direction of the boundary layer varies between
–5/3 and –1. Thus, at the top of the boundary layer, the assumption of local homogeneity and isotropy is
not satisfied (Xian et al., 2024b). Therefore, when calculating the dissipation rate, we consider $n$ as a
variable and obtain it by linearly fitting the turbulence spectrum in logarithmic coordinates (Xian et al.,
2024b), which is then substituted into Equation (4) to obtain the dissipation rate. Figure 4 shows a
comparison of TKE dissipation rates obtained with the three-dimensional ultrasonic anemometer (shown
in blue) and the wind lidar (shown in orange) from October 1 to 9, 2022. It can be seen that the two data
sets have high consistency.

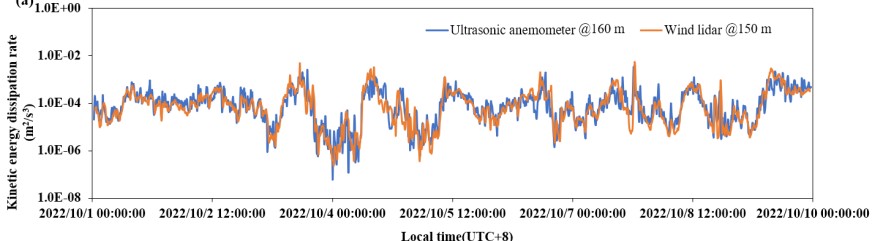

**Figure 4.** Comparison of the TKE dissipation rate obtained with the ultrasonic anemometer (blue) and wind lidar
(orange).

### 3.4 Determination of the Shear Generation Term

Shear generation is a key term in the turbulent energy budget, representing the generation of
turbulent energy due to the vertical shear of the wind (i.e., the variation of wind speed with height). This
term is calculated by multiplying the turbulent shear stress, $u'w'$ ($v'w'$), by the wind speed gradient. In this
study, the shear generation term of the wind lidar is calculated by the following steps.

(1) *Calculate the turbulent shear stress*. The shear generation term is the product of turbulent
fluctuations along the wind direction and vertical wind speed components, i.e. $u'w'$ and $v'w'$.

(2) *Calculate the shear generation term*. This is done using the following equation

$$S = -\overline{u'w'}\frac{\Delta u}{\Delta z} - \overline{v'w'}\frac{\Delta v}{\Delta z} . \tag{5}$$

For wind lidar, due to its spatial resolution of 30 m, $\Delta z = 30$ m. By calculating the shear generation terms
for different height layers, the shear generation terms for the entire boundary layer are obtained.

Figure 5 shows a comparison of the turbulent shear stresses, $u'w'$ and $v'w'$, measured by the ultrasonic
anemometer (in blue) and the wind lidar instrument (in orange) from October 1 to 9, 2022. It can be seen
from this figure that the wind lidar can reflect the trend of turbulent shear stress very accurately.



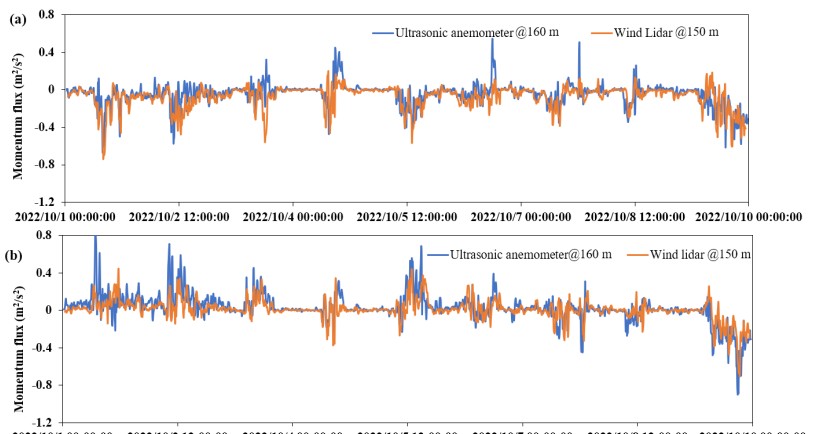

**Figure 5.** Comparison of the turbulent shear stresses, $u'w'$ and $v'w'$, obtained with the ultrasonic anemometer (blue) and wind lidar (orange).

### 3.5 Determination of the Turbulent Transport Term

The turbulent transport term is calculated by calculating the vertical derivative of the product of the turbulent velocity component in the vertical direction and TKE, as shown in the formula

$$T_t = -\frac{\partial w' E'}{\partial z} = -\frac{1}{2}\frac{\Delta(w'u'^2 + w'v'^2 + w'w'^2)}{\Delta z}. \tag{6}$$

As can be seen, it is necessary to calculate the third-order moments of the vertical and horizontal wind speed components for each altitude layer, namely $w'u'^2$, $w'v'^2$, and $w'w'^2$. Furthermore, the turbulent transport term can be estimated by taking the spatial derivatives of the aforementioned third-order moments. For the wind lidar, due to its spatial resolution of 30 m, $\Delta z = 30$ m. For comparison, we performed spatial differentiation on the data obtained with the ultrasonic anemometer at 160 m and 320 m to obtain a slightly rough transmission term, which was compared with the turbulent transmission term obtained from the wind lidar, as shown in Figure 6. From the graph, it can be seen that although there is a distance of 160 m between the two ultrasonic anemometers (shown in blue), the results obtained are still consistent with those of the wind lidar (shown in orange), which to some extent indicates that the wind lidar can reflect the trend of the turbulence transport term.

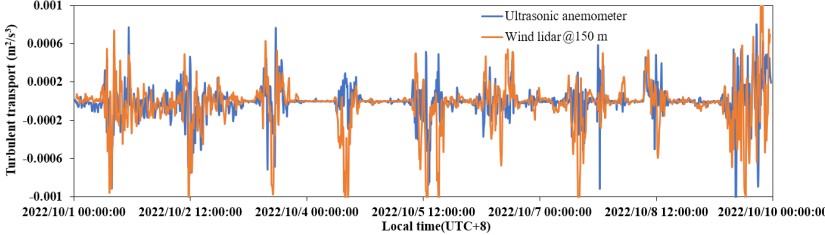

**Figure 6.** Comparison of turbulent transport terms obtained with the ultrasonic anemometer (blue) and wind lidar (orange).



### 3.6 Determination of the Pressure Transport Term

Static pressure fluctuations in the atmosphere are very small (0.01–0.05 hPa), and hence, are extremely difficult to measure, and there is little understanding of this term currently. Previous studies have attempted to use micro barometers and vertical displacement sonar anemometers to calculate the pressure velocity covariance, but the results obtained were very scattered and did not show a clear trend towards stationarity (Pozzobon et al., 2023; Acevedo et al., 2016). In practical applications, researchers

have used other methods to address this challenge. In some cases, the pressure transport term is estimated through residual calculations, which indicate that it negligible in practical operations; therefore, it is ignored in this study.

### 3.7 Determination of the Buoyancy Generation Term

From the composition of the buoyancy generation term, it can be seen the calculation of $B$ requires

high-resolution temperature profile data. However, at present, due to the lack of vertical detection methods with high spatial and temporal resolution for temperature, vertical structural detection of buoyancy generation terms remains a major technical challenge. Therefore, in this study, we obtain the $B$ term indirectly. After obtaining each budget term earlier, the buoyancy generation term was obtained through the residual method, i.e.,

$$B = E_t - S - T_t + D .$$
(7)

The accuracy and reliability of the $B$ term will directly affect the reliability and accuracy of the calculated results of the other terms. Due to the ability of wind lidar to obtain accurate three-dimensional wind speeds, the terms $Et$, $S$, and $T_t$ are accurately obtained in turn. Therefore, the error mainly comes from the calculation of the $D$ term and the assumption that the pressure transport term, $T_p$, is negligible.

We will test this assumption in Section 4.

### 4 Results and Discussion

Based on the proposed method, we obtained the distribution of the buoyancy generation term at different heights/layers within the boundary layer. According to the equation for the buoyancy generation term, the data from the three-dimensional ultrasonic anemometer and thermometer can be inserted into

it to obtain the buoyancy generation terms for the two altitude layers for which the data were obtained.

A comparison of the lidar data obtained at 150 m and 330 m with the ultrasonic anemometer data measured at 160 m and 320 m is shown in Figure 7. From the figure, it can be seen that there is a high degree of consistency between the two altitude levels, which preliminarily verifies that the wind lidar instrument can achieve vertical monitoring of the buoyancy generation term. Furthermore, based on the

buoyancy generation term gleamed from the three-dimensional ultrasonic anemometer data, the errors of the wind lidar data were calculated, as shown in Figures 8(a) and (b). From this error distribution, it can be inferred that the average error is very close to 0, indicating that the proposed method does not have significant systematic errors. According to the statistical results, at a height of 160 m, 48% of the results have an error of less than 0.0001 $m^2/s^3$; at 320 m, 47% of the results have an error of less than 0.0001

$m^2/s^3$. After using the 3σ method to remove abnormal data, correlation graphs of the buoyancy generation



term obtained from the three-dimensional ultrasonic anemometer and wind lidar data were plotted, as shown in Figure 8(c) and (d), respectively. From the plot, it can be seen that the correlation between the two data sets is greater than 0.9 at both heights, indicating that the wind lidar data have very high reliability.

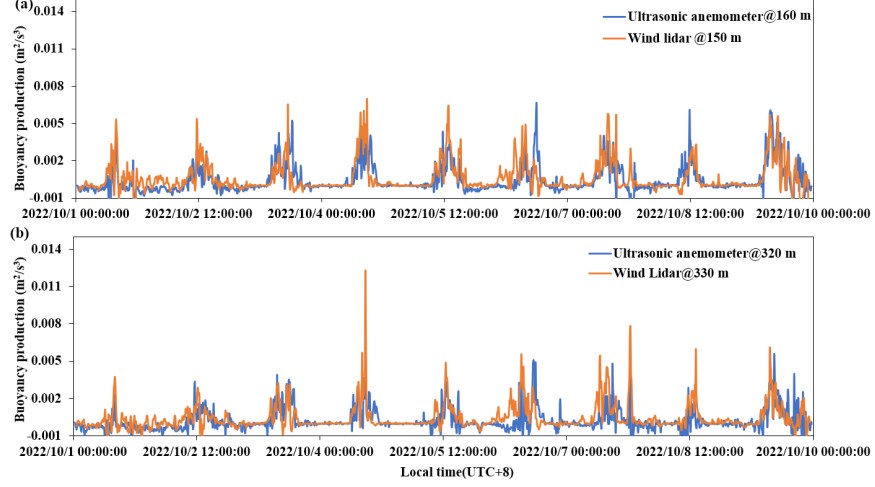

**Figure 7.** Comparison of buoyancy generation terms obtained with the ultrasonic anemometer (blue) and wind lidar (orange).

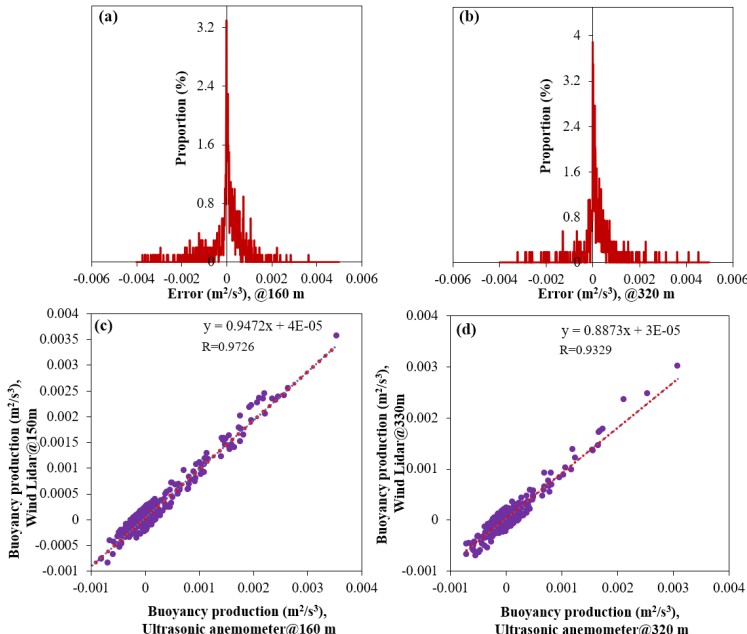

**Figure 8.** Correlation and error analysis of buoyancy generation terms obtained with the wind lidar (top row) and ultrasonic anemometer (bottom row).

Through the previous series of comparisons, it was verified that the method proposed in this study can effectively detect the various budget terms of TKE. To observe the vertical structural changes of



each budget term, we further selected several examples of different weather conditions for analysis, namely a cloudy and light rainy day, a sunny day with cumulus clouds, and a sunny day with low-level
jets at night.

Figure 9 presents the spatiotemporal distribution of horizontal wind speed (a), vertical wind speed (b), TKE (c), turbulent trend (d), turbulent transport term (e), dissipation rate (f), shear generation term (g), and buoyancy generation term (h) on October 1, 2022. The weather on that day consisted of low clouds with light rain (0.1 mm) around 17:00. The lowest temperature was 26°C and the highest
temperature was 29°C. From the horizontal wind speeds shown in Figure 9(a), it can be seen that there were low-level jets throughout the day, with wind speeds ranging from 10–15 m/s at an altitude of approximately 700 m. During the period from 00:00 to 08:00 local time, it can be seen from Figure 9(g) that the shear generation term caused by the horizontal wind had a relatively large value. However, due to the negative buoyancy generation term (as shown in Figure 9(h)), it did not have an increasing effect
on the turbulence overall. Therefore, the results during this period correspond to the phenomenon of weak TKE from 00:00 to 08:00 (Figure 9(c)). From 08:00 to 12:00, the ground was affected by the increase in daytime solar radiation, and the buoyancy generation term started to become positive, playing a role in generating TKE. Combined with the shear generation term generated by low-level jets, the TKE reached its maximum throughout the day during this period, as shown in Figure 9(c). Afterward, due to
the cover of low clouds, there was less ground radiation, and the buoyancy generation term was basically negative, which mainly suppressed and dissipated turbulence. However, the shear generation term caused by the still existing low-level jets had a relatively large value, occupying the main guiding role in the generation and maintenance of turbulent energy, resulting in strong TKE.

Figure 10 shows the vertical distribution profiles of each budget term during the day (13:00) and at
night (22:00). From Figure 10(a), it can be seen that below 400 m, the turbulent transport term was generally positive, consistent with the positive turbulent transport term observed by (Nilsson et al., 2016a) under cloud cover. The difference is that our method can intuitively reveal the variation of the entire turbulent transport term with height; for example, in the height range of 400 m to 1000 m, it was about zero. From Figure 10(b), it can be seen that below 800 m, the dissipation rate was relatively small, and
the buoyancy generation term was mainly negative, corresponding to a positive shear generation term, which was slightly greater than the buoyancy generation term. Therefore, turbulence was still increasing, corresponding to the TKE in Figure 9(c) still being active at night. Unlike previous tower-based observations (Nilsson et al., 2016a), we can accurately determine the height range where the buoyancy generation term is negative. For example, in Figure 10(b), the area where the buoyancy generation term
is negative is below 780 m. From this example, we can see that the method proposed in this study can effectively reflect the dissipation effect of buoyancy generation on TKE under low cloud and light rain conditions.





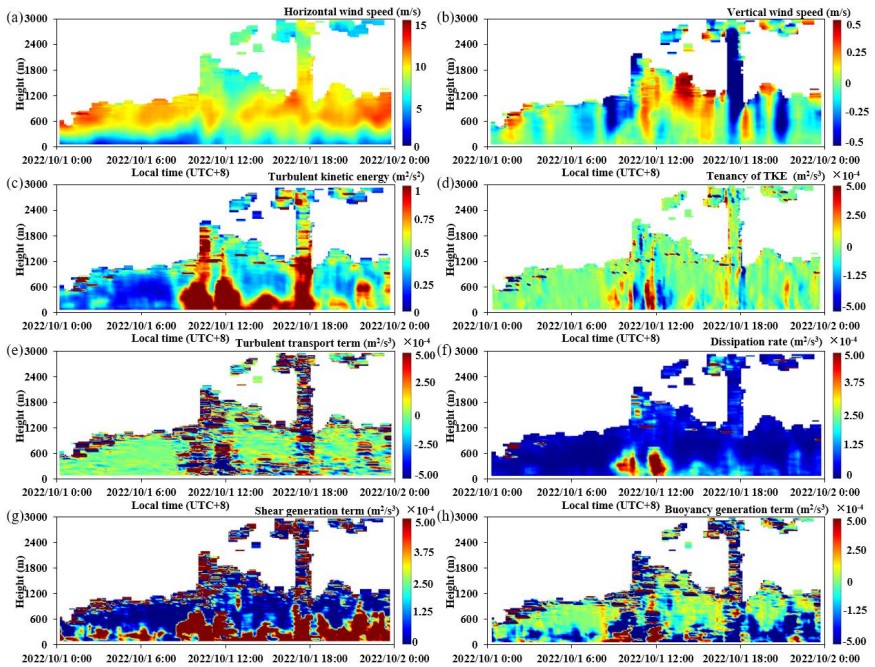

**Figure 9.** Temporal and spatial distributions of the horizontal wind speed (a), vertical wind speed (b), TKE (c), turbulent trend (d), turbulent transport term (e), dissipation rate (f), shear generation term (g), and buoyancy generation term (h) on October 1, 2022 (a cloudy and light rain day).

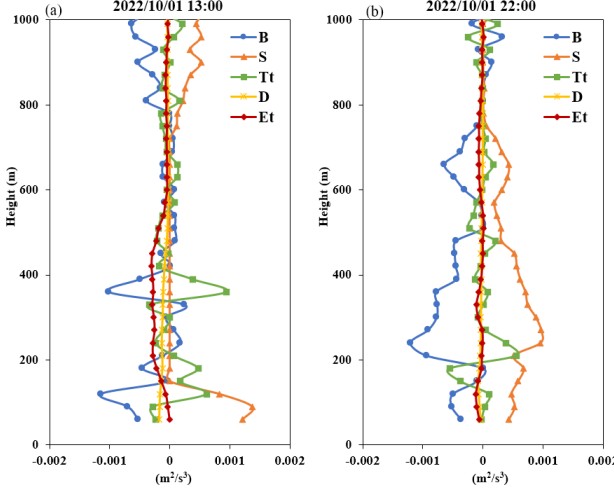

**Figure 10.** Vertical distribution profile of the budget terms for daytime (13:00) and nighttime (22:00) on October 1, 2022.


Figure 11 shows the spatiotemporal distributions of the horizontal wind speed (a), vertical wind speed (b), TKE (c), turbulent trend (d), turbulent transport term (e), dissipation rate (f), shear generation term (g), and buoyancy generation term (h) on October 5, 2022. The day was sunny with cumulus clouds,



with a minimum temperature of 27°C and a maximum temperature of 32°C. From Figures 11 (f), (g),
and (h), it can be seen that from 00:00 to 08:00, the shear and buoyancy generation terms were basically
zero, and the dissipation rate was relatively small, which had no effect on the generation and dissipation
of turbulence, corresponding to the weaker TKE seen in Figure 11(c). From 08:00 to 18:00, the shear
and buoyancy generation terms were generally positive during the day. But the value of the buoyancy
generation term was larger than that of the shear generation term, reaching a maximum of 1.8 km in the
vertical direction, and playing a dominant role in the generation and maintenance of turbulent energy
throughout the boundary layer. After 18:00, the buoyancy generation term and shear generation term
were basically zero, corresponding to the weaker TKE shown in Figure 11(f).

Figure 12 shows the vertical distribution profile of each budget term during the day (13:00) and at
night (22:00). From Figure 12(a), it can be seen that turbulent transport was the main dissipation term,
which means that the turbulence generated near the surface was transported to the surrounding
environment and the upper part of the boundary layer. Buoyancy was the main generation term, which
was approximately one order of magnitude larger than the turbulent transport term. Therefore, turbulence
was generally increasing, consistent with the situation where the buoyancy generation term was dominant
in sunny convective conditions. From Figure 12(b), it can be seen that the buoyancy generation term was
mainly negative, and the shear generation term was positive, both of which were of equal magnitude,
reflecting the suppression of nighttime turbulence. In this example, we can see that the turbulent transport
term during a clear and cloudless daytime could have a dissipative effect on TKE.

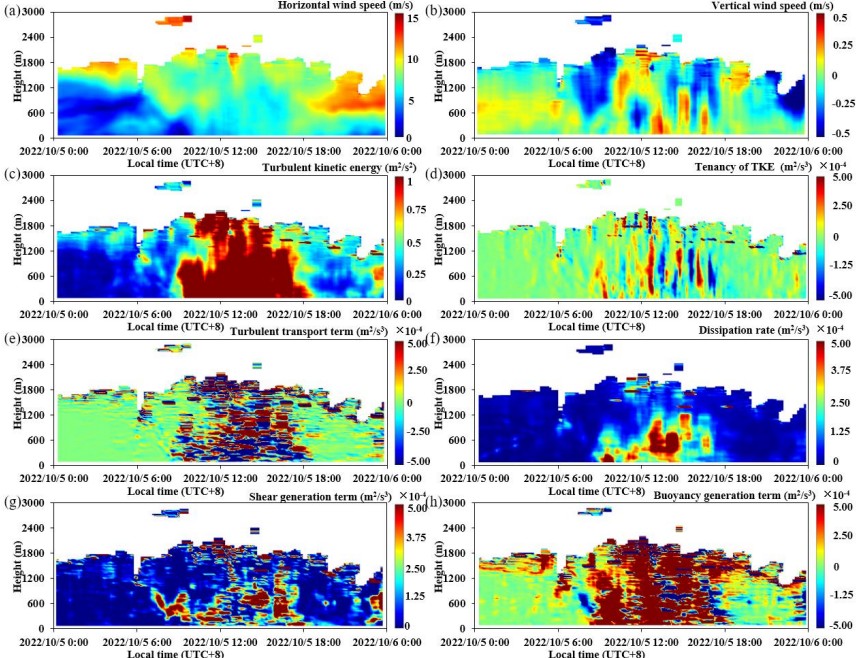

**Figure 11.** Temporal and spatial distributions of the horizontal wind speed (a), vertical wind speed (b), TKE (c),
turbulent trend (d), turbulent transport term (e), dissipation rate (f), shear generation term (g), and buoyancy
generation term (h) on October 5, 2022 (a sunny day with cumulus clouds).





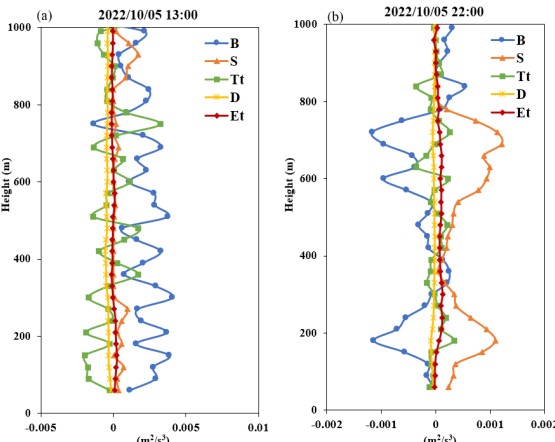

**Figure 12.** Vertical distribution profile of the budget terms for the daytime (13:00) and nighttime (22:00) on October
5, 2022.

Figure 13 shows the spatiotemporal distributions of the horizontal wind speed (a), vertical wind speed (b), TKE (c), turbulent trend (d), turbulent transport term (e), dissipation rate (f), shear generation term (g), and buoyancy generation term (h) on October 9, 2022. The weather on that day was sunny, with a minimum temperature of 22°C and a maximum temperature of 30°C. At 08:00, the wind speed was relatively small, and due to the fact that the buoyancy generation term was basically zero before sunrise (as shown in Figure 13(h)), the TKE was relatively small, and the turbulence was relatively weak during this period, as shown in Figure 13(c). At 08:18, the ground became heated by solar radiation, and the buoyancy generation term was positive, which developed vertically to about 1 km, generating and maintaining turbulence. Corresponding to this period in Figure 13(c), the TKE was the highest throughout the day. After 18:00, the buoyancy generation term became negative, suppressing the generation and maintenance of turbulence. However, at the same time, low-level jets began to appear, with maximum wind speeds greater than 15 m/s, generating a larger shear generation term that offset the inhibitory effect of the buoyancy generation term and strengthened turbulent motion. Corresponding to Figure 13(c), it can be seen that the value of TKE was still relatively large during this period.

Figure 14 shows the vertical distribution profile of each budget term during the day (13:00) and at night (22:00). From Figure 14(a), it can be seen that the shear generation term was about zero, with the dissipation rate and turbulence transport being the main dissipation terms, and buoyancy being the main generation term; the latter's magnitude was about twice that of the turbulence transport term, so turbulence always intensified. From Figure 14(b), it can be seen that the buoyancy generation term and turbulent transport term exhibited similar magnitudes, which basically cancelled each other out at both heights. In addition, shear was the main generation term, which caused the turbulence, and hence turbulent energy, to increase, as shown in Figure 13 (c). From this example, we can see that, by using the method proposed in this study, we have determined that the main dissipation effect during a clear and cloudless day primarily comes from the dissipation rate term, while the shear generation term under low-level jets enhances the turbulent energy, during clear and days and nights.





In summary, from the individual cases of different weather conditions mentioned above, we can see that during the day, turbulent transport and the dissipation rate are the main dissipation terms, while buoyancy generation is the main dissipation term at night. In addition, with the proposed method we are able to monitor the vertical distribution of various budget terms within the boundary layer, and can comprehensively reflect the impact of each budget term on the vertical structure of turbulent energy.

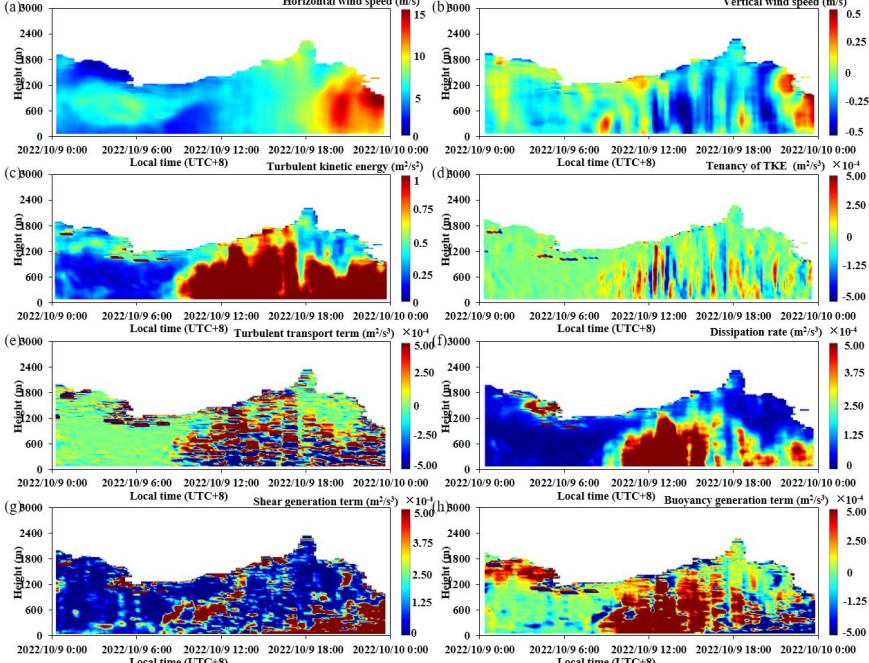

**Figure 13.** Temporal and spatial distributions of the horizontal wind speed (a), vertical wind speed (b), TKE (c), turbulent trend (d), turbulent transport term (e), dissipation rate (f), shear generation term (g), and buoyancy generation term (h) on October 9, 2022 (a sunny day with low-level jets at night).

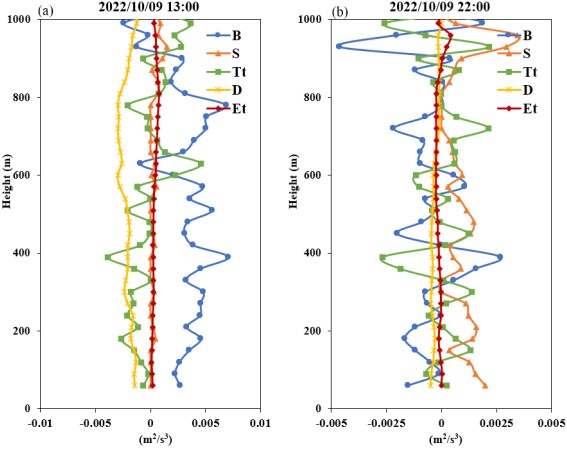



**Figure 14.** Vertical distribution profile of the budget terms for daytime (13:00) and nighttime (22:00) on October 9, 2022.

**Conclusion**

We proposed a new method for detecting turbulent energy budget terms based on coherent wind lidar, and through strict data quality control, ensured the accuracy and reliability of the obtained data. By comparing these data with those obtained with a three-dimensional ultrasonic anemometer, it was shown that the errors are relatively small, and are less than 0.0001 $m^2/s^3$ for at least 47% of the data, which

verify the accuracy and reliability of our method. This study explored the generation and dissipation mechanisms of turbulence under different weather conditions, and analyzed the effects of shear generation, buoyancy generation, turbulence transport, and dissipation rate on turbulent motion for different periods throughout the day and night. The results indicate that the buoyancy generation term plays a dissipative role in TKE under low cloud and light rain conditions. During the day, turbulent

transport and the dissipation rate are the main dissipation terms, while buoyancy generation is the main dissipation term at night. The results show that the proposed method can accurately capture key parameters such as TKE, dissipation rate, shear generation, turbulent transport, and buoyancy generation, and accurately describe the spatiotemporal distribution characteristics of each budget term of turbulent energy. These research results not only provide new a perspective and method for studies of atmospheric

turbulence, but can also be further applied to a wider range of meteorological research topics and practical applications, especially those related to fine observations of the vertical structure and dynamics of turbulence.

**Data availability**

The data are available from the authors upon request.

**Author contributions**

Conceptualization: J.X. and H.Y.; methodology: J.X. and N.Z.; software: J.X.; validation: Y.H. and Z. Q.; formal analysis: J.X. and N.Z.; investigation: H.Y and Z. Q.; resources: C.L. and H. L.; data curation: Y.Y. and H. L.; writing—original draft preparation: J.X.; writing—review and editing: H.Y. and N.Z.; visualization: X.L.; supervision: H.Y. and N.Z.; project administration: H.Y. and N.Z.; and funding

acquisition: H.Y. and N.Z.

**Competing interests**

The contact author has declared that none of the authors has any competing interests.

**Acknowledgements**

We thank Shenzhen Darsunlaser Technology Co., Ltd. and Xinlin Yang.



**Financial support**

This work was supported by the Special Project for Sustainable Development of Shenzhen (KCXFZ20201221173412035), the National Natural Science Foundation of China (NSFC; U2342221 and 42275065), Guangdong Province Science and Technology Department Project (2021B1212050024), Scientific research projects of Guangdong Provincial Meteorological Bureau (GRMC2020M29), Science

and Technology Innovation Team Plan of Guangdong Meteorological Bureau (GRM-CTD202003), Open Fund for the Key Open Laboratory of Urban Meteorology of the China Meteorological Administration (LUM-2024-03), and China Postdoctoral Science Foundation (2024M751378).

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
