# Peer review of "Turbulent Energy Budget Analysis Based on Coherent Wind Lidar Observations"

_EGUsphere, 2024_

## Author Comment (AC1)

**Egusphere-2024-2163- Response Letter 1**

Dear Editor and reviewers,

We would like to thank the reviewers and editor for their comments that have allowed us to further clarify some aspects of the manuscript in this revised version. Hereafter, we report reviewers' comments and our replies (*in italics*). For yours and reviewers' convenience we have put the corresponding major changes introduced in red color in the revised version of the manuscript.

**Reviewer 1:**

This manuscript presents a new method for obtaining turbulent kinetic energy (TKE) budget estimates from wind lidar measurements. The budget term estimates are compared to values based on ultrasonic anemometer data. The main benefit of the proposed new method is that it can provide values of the different terms of the TKE budget throughout the vertical column and not limited to point measurements such as from the ultrasonic anemometer. The manuscript is well written and easy to follow, with a clear structure and mostly clear figures.

My main concern with this manuscript is how the accuracy of the new lidar-based method compared to the established ultrasonic anemometer-based method is judged and communicated. In general, the basis on which the authors conclude that the method is accurate is somewhat vague and I would like to see more critical discussion of the results.

**On multiple occasions (such as on lines 189, 219, and 234) it is stated that the lidar data is consistent with the ultrasonic anemometer data but this claim is vague; is it possible to quantify the agreement?**

Response: *As the reviewer suggests, we have added correlation coefficients to quantify consistency in the comparison of wind lidar and ultrasonic anemometer results. For example, in Figure 3, it can be seen that the two data sets have a very high degree of consistency, with a correlation coefficient of 0.98; in Figure 3(b), it can be seen from this plot that the wind lidar data can reflect the trend of turbulence changes very well, with a correlation coefficient of 0.97; in Figure 4, the consistency between the two is relatively good, with a correlation coefficient of 0.96.*

**And how do the authors view the discrepancies that do exist? For example, in Figure 5(a) there are multiple positive peaks in the momentum flux from the ultrasonic anemometer that the wind lidar does not capture, why is that? Please include some discussion of if there are any conditions under which this method may not be suitable to use (as well as some comment on possible deficiencies of the ultrasonic anemometer data, while that is a more established method it is not the "truth").**

Response: *Thanks for the reviewer's professional comments. There are multiple positive peaks in the momentum flux detected by the ultrasonic anemometer, but the wind lidar did not observe the same phenomenon at the corresponding time. It is speculated that one of the reasons is that the spatial resolution of the wind lidar is 30 m, which means that its results are the average effect within these 30 m, equivalent to smoothing the data, resulting in a difference in peak size between the two. Another possibility is that ultrasonic anemometers have a higher monitoring frequency (10 Hz), compared to wind lidar's 0.2 Hz, which can detect faster changes in high-frequency energy. As the reviewer suggests, we have added the texts in revised version. (See lines 247 to 254)*

*Based on the detection principle of wind lidar, the method proposed in this study is applicable during sunny and cloudy conditions; however, it is not suitable for deployment during periods of heavy rainfall. Furthermore, this method holds potential for elucidating turbulence convection interactions and convective initiation before precipitation occurs. It is imperative to acknowledge that due to the inability to measure pressure transport terms*

*and monitor high-frequency turbulent energy, the error of the proposed method may increase in weather processes dominated by these two factors As the reviewer suggests, we have added the texts in the conclusions. (See lines 447 to 451)*

**Furthermore, in the abstract and conclusions it is emphasized that the errors are "less than 0.0001 m$^2$/s$^3$, for at least 47% of the data", which to me does not sound very convincing as it leaves the possibility that the remaining 53% of the data could have large errors. Figure 8 shows that this is not the case, but I recommend that the authors comment on this both in the abstract and in the main body of the text. In the abstract and conclusions it should be made clear that those numbers pertain to the buoyancy term as this is currently not mentioned.**

Response: *Thanks for the reviewer's professional comments. By comparing these data with those obtained with a three-dimensional ultrasonic anemometer, the results indicate that the error of the buoyancy generation term detected by the proposed method is relatively small, with an average absolute value of less than 0.00014 m$^2$/s$^3$, which verify the accuracy and reliability of our method. As the reviewer suggests. we have added the texts in the abstract and conclusions. (See lines 444 to 451)*

**Overall, I find this an interesting paper and the proposed method provides information that is useful for understanding the generation and dissipation of turbulence in different atmospheric conditions, and with the inclusion of a more rigorous discussion of the validity of the results as well as consideration of the specific comments below I find it suitable for publication.**

Response: *Thanks for the reviewer's professional comments.*

**Line by line comments:**

**1.Line 11: "The turbulent kinetic energy (TKE) budget term, as a key physical quantity […]" The subsequent sentences discuss budget terms in plural, should it be plural here as well? The use of the words "TKE budget term" is used at multiple occasions throughout the manuscript, please make it clear if you are referring to a specific term in the budget or the budget as a whole (for example on line 32).**

Response: *Thanks for the reviewer's comment. It should be "TKE budget terms". We have modified the texts in revised version. (See line 11)*

**2.Line 38: "(YSU, MYJ, MYNN2, ACM2, etc.)" These acronyms should be defined and ideally references for the schemes should be provided.**

Response: *As the reviewer suggests, we have modified the texts in revised version. (See lines 38 to 41)*

**3.Line 107: What does "strong representativeness" mean? Please elaborate on what the data is representative of.**

Response: *As the reviewer suggests, we have modified the texts in revised version. The wind field and temperature data obtained by the three-dimensional ultrasonic anemometer and thermometer on the gradient observation tower can represent the environmental characteristics of the region. (See lines 108 to 111)*

**4.Line 134: When it says that "the wind speed measurements are checked every 30 min every day", what is involved in the "checking"?**

Response: *Thanks for the reviewer's comment. We calculate the average and standard deviation of wind speed and direction every 30 minutes, and use the triple standard deviation principle to remove outliers. We have made revisions in the text for better readability. (See lines 138 to 139)*

**5.Line 143: Is "[29]" a literature reference? If so, please write it in the same format as the other references. If not, please explain what it means.**

Response: *Thanks for the reviewer's comment. "[29]" is a literature reference. We have updated the citation of this literature. (See lines 148 to 149)*

**6.Figure 2: In this figure it is very difficult to judge the agreement between the blue and orange lines since they mostly overlap such that the orange line covers the blue one. Consider plotting the difference between the two lines (or some other measure of the difference) rather than the absolute values to facilitate comparison. To some extent this is true also for the other figures that compare two timeseries, but the problem is the biggest in Figure 2.**

Response: *As the reviewer suggests, we have updated the Figure 2 in revised version.*

**7.Line 174: θv usually denotes virtual potential temperature (so also in Stull, 1988 and Nilsson et al., 2016a which are referenced); it makes it easier for the reader if conventions for variable naming are kept.**

Response: *As the reviewer suggests, we have modified the texts in revised version.*

**8.Line 176: I believe it should be "tendency" rather than "tenacy"; also on Line 188 and in the caption to figure 3.**

Response: *As the reviewer suggests, we have modified the texts in revised version. (See lines 181 to 185)*

**9.Line 185: The text says that Figure 3(a) shows data from the wind lidar obtained at both 150 and 160 m height but the figure seems to show only data from 150 m.**

Response: *Thanks for the reviewer's comment. We have modified the texts in revised version. Due to the spatial resolution of the wind lidar data being 30 m, the TKE at a height of 150 m was selected from the vertical profile obtained by wind lidar (shown in orange) and compared with the results obtained by the three-dimensional ultrasonic anemometer at a height of 160 m (shown in blue) on the tower from October 1 to 9, 2022, as shown in Figure 3(a). (See lines 190 to 193)*

**10.Figure 5: What is the value for delta z for the ultrasonic anemometer?**

Response: *Thanks for the reviewer's comment. Ultrasonic anemometers were installed on the tower at heights of 160 m and 320 m. For the ultrasonic anemometer, $\Delta z = 160$ m. We have added the texts in revised version. (See line 241)*

11.Line 266: Please provide a reference to the statement that the pressure transport tern is negligible in practical operations.

Response: *As the reviewer suggests, we have added citations in revised version. "In some cases, the pressure transport term is estimated through residual calculations, which indicate that it negligible in practical operations (Kaimal and Finnigan, 1994; Wyngaard, 2010; Pozzobon et al., 2023); therefore, it is ignored in this study." (See lines 280 to 282)*

12.Line 276: "affect" should be "be affected by", since B is derived as the residual and thus cannot affect the accuracy of the other terms.

Response: *As the reviewer suggests, we have modified the texts in revised version. (See line 291)*

13.Line 290: "gleamed" -> "gleaned"

Response: *As the reviewer suggests, we have modified the texts in revised version. (See line 305)*

14.Line 289-291: Please describe how the errors were calculated (what definition of the error is used).

Response: *As the reviewer suggests, we have modified the texts in revised version. The buoyancy generation term ($B'$) gleaned from the three-dimensional ultrasonic anemometer data was used as the standard value. The error ($\Delta B = B - B'$) of the buoyancy generation term (B) detected by the wind lidar was calculated, and its distribution was statistically analyzed, as shown in Figures 8(a) and (b). (See lines 304 to 307)*

15.Figure 8 caption: I find that this caption does not accurately describe the contents of this figure.

Response: *As the reviewer suggests, we have modified the caption in revised version.*

16.Paragraph starting with Line 311: The period after 12:00 is not mentioned in the text, please provide some comment on the interpretation of that part of the figure.

Response: *As the reviewer suggests, we have added the texts in revised version. From 12:00 to 19:00, due to the cover of low clouds, there was less ground radiation, and the buoyancy generation term was basically negative, which mainly suppressed and dissipated turbulence. However, the shear generation term caused by the still existing low-level jets had a relatively large value, occupying the main guiding role in the generation and maintenance of turbulent energy, resulting in strong TKE. After 19:00, although the shear generation term remained relatively large, the dissipation effect of the buoyancy generation term also increased, leading to a weakening of TKE. (See lines 344 to 350)*

17.Figure 9: Since all the data is from the same day the date (2022/10/1) can be removed from the x-axis labels, this would make the hours on the axis easier to read (the same is true for figures 11 and 13).

Response: *As the reviewer suggests, we have updated the figures in revised version.*

**18.Figure 9: Please emphasize somehow that the range of the color scales differs between the panels (and potentially change the color scales such that zero has the same color in all panels).**

Response: *As the reviewer suggests, we have modified the texts in revised version. The color scale range of panels varies for different TKE budget items, but for the same TKE budget item, the color scale range remains unchanged. (See lines 330 to 332)*

**19.Line 331: "observed by (Nilsson et al., 2016a)" -> "observed by Nilsson et al. (2016a)"**

Response: *As the reviewer suggests, we have modified the texts in revised version. (See line 352)*

**20.Figure 10: A thin vertical indicating zero would be helpful for seeing if the terms are positive and negative.**

Response: *As the reviewer suggests, we have updated the figures in revised version.*

**21.Data availability statement: According to the ACP data policy, "The best way to provide access to data is by depositing them (as well as related metadata) in FAIR-aligned reliable public data repositories, assigning digital object identifiers, and properly citing data sets as individual contributions". Providing the data in an easy to access format, in a public repository, is much preferable to providing it only upon request.**

Response: *As the reviewer suggests, we have stored the dataset on Github and generated a DOI through Zenodo. The wind field data measured by ultrasonic anemometer can be downloaded from the Shenzhen Data Open Platform (https://opendata.sz.gov.cn/data/dataSet/toDataDetails/29200_00900273). Data to generate the figures of this paper are available at https://doi.org/10.5281/zenodo.13624484(Xian et al., 2024a). (See lines 465 to 468)*

On behalf of all authors,
Sincerely,
Honglong Yang

Shenzhen National Climate Observatory
Meteorological Bureau of Shenzhen Municipality
518000 Shenzhen, China
E-mail: yanghl01@163.com

---

## Author Comment (AC2)

**Egusphere-2024-2163- Response Letter 2**

Dear Editor and reviewers,

We would like to thank the reviewers and editor for their comments that have allowed us to further clarify some aspects of the manuscript in this revised version. Hereafter, we report reviewers' comments and our replies (*in italics*). For yours and reviewers' convenience we have put the corresponding major changes introduced in red color in the revised version of the manuscript.

**Reviewer 2:**

A scientifically sound turbulent energy budget analysis is required for better understanding of the generation and dissipation processes of turbulence. However, current research on the generation and dissipation mechanisms of atmospheric turbulence energy is mainly based on ground or tower base observations, leading to unknown vertical TKE budget term. The authors propose a new method based on coherent wind lidar to detect TKE budget terms and compare them with data from a three-dimensional ultrasonic anemometer for verification. The results indicate that their proposed method can comprehensively reflect the impact of each budget term on the vertical structure of TKE, providing a new perspective and method for atmospheric turbulence research. The expression of this paper is clear, the argument is reasonable. It is suitable for publication. I think there are some small issues that can be improved, which is shown as follows:

Response: *Thanks for the reviewer's professional comments.*

**Minor comments:**

**1.Lines 176 and 188 :"tenacy" should be "tendency".**

Response: *As the reviewer suggests, we have modified the texts in revised version. (See lines 181 to 185)*

**2.In section 3.7 Determination of the Buoyancy Generation Term: I suggest the authors elaborate on the sources of errors.**

Response: *As the reviewer suggests, we have modified the texts in revised version. Due to the ability of wind lidar to obtain accurate three-dimensional wind speeds, the terms Et, S, D, and Tt are accurately obtained in turn. Therefore, the error mainly comes from the assumption that the pressure transport term, Tp, is negligible. (See lines 293 to 295)*

**3.Figure 8 shows that at the height of 160 m, 48% of the results have an error of less than 0.0001 m2/s3; At the height of 320 m, 47% of the results have an error of less than 0.0001 m2/s3. The error statistical method is not rigorous enough and should be given as mean error or standard deviation.**

Response: *Thanks for the reviewer's professional comments. By comparing these data with those obtained with a three-dimensional ultrasonic anemometer, the results indicate that the error of the buoyancy generation term detected by the proposed method is relatively small, with an average absolute value of less than $0.00014 \ m^2/s^3$, which verify the accuracy and reliability of our method. As the reviewer suggests. we have added the texts in the abstract and conclusions. (See lines 444 to 447)*

**4.The caption of Figure 8 does not effectively convey the meaning of this figure.**

Response: *As the reviewer suggests, we have modified the caption in revised version.*

**5.Lines 293 and 294: How do you calculate the error? The calculation method for the error should be provided in the text.**

Response: *As the reviewer suggests, we have modified the texts in revised version. The buoyancy generation term ($B'$) gleaned from the three-dimensional ultrasonic anemometer data was used as the standard value. The error ($\Delta B = B - B'$) of the buoyancy generation term ($B$) detected by the wind lidar was calculated, and its distribution was statistically analyzed, as shown in Figures 8(a) and (b). (See lines 304 to 307)*

**6.Can this method proposed here be applicable in other circumstances? e.g., How about the implications for elucidating the turbulence-convection interaction, and convection initiation.**

Response: *As the reviewer suggests, we have added the texts in the conclusions. Based on the detection principle of wind lidar, the method proposed in this study is applicable during sunny and cloudy conditions; however, it is not suitable for deployment during periods of heavy rainfall. Furthermore, this method holds potential for elucidating turbulence convection interactions and convective initiation before precipitation occurs. It is imperative to acknowledge that due to the inability to measure pressure transport terms and monitor high-frequency turbulent energy, the error of the proposed method may increase in weather processes dominated by these two factors. (See lines 447 to 451)*

**7.Lines 53-56: "...including changes in surface heat flux, atmospheric stability, and topography". More recent references are needed to support this statement. The authors can refer to https://doi.org/10.5194/acp-21-17079-2021.**

Response: *As the reviewer suggests, we have added some relevant references in revised version.*

**8.Lines 58-59: Radar wind profiler can provide such high-resolution turbulence measurements (doi:10.1016/j.uclim.2022.101151), and can be mentioned here..**

Response: *As the reviewer suggests, we have added some relevant references in revised version.*

On behalf of all authors,
Sincerely,
Honglong Yang

Shenzhen National Climate Observatory
Meteorological Bureau of Shenzhen Municipality
518000 Shenzhen, China
E-mail: yanghl01@163.com

---

## Editor Decision (ED1)

In this paper the authors use a wind lidar to derive the terms in the TKE-budget. The two reviewers both had objections to some of the authors text and explanations and the authors responded to these in an adequate fashion. I have now reviewed the paper and I have some major concerns that I want to have answered before finally deciding on this paper.

*Major comments*

I am a big fan of using remote sensing instruments for this type of study and I think the time-height distributions shown in the last section of the paper illustrates their potential. There are, however, two potential major flaws in the representation of the TKE budget terms, especially relating to the dissipation term and the buoyancy term. Most of the comments below are based on the authors not properly explaining the choice of parameters in their Reynolds averaging for the two different instruments. I would also point out that with one of the termsin the budget equation, the dissipation, being estimated indirectly and another being calculated as a residual, the average budget profiles, like in Figure 10 & 12 must balance to zero; hence, if the dissipation is larger than the shear production, the buoyancy has to pick up the rest to balance the results. Therefore, Figure 12a is puzzling since that doesn't seem to happen here.

Starting with the buoyancy term, this is - contrary to the authors claims - *never measured*; it is estimated as a residual – after ignoring the pressure transport term which in my opinion *is* OK. But the reason to believe in the method comes down to the residual – assumed to be buoyancy – being much larger compared to other leading terms; that the residual cannot be explained in terms of errors in the other terms.

One could have assumed the time tendency term to be small and then proceed to calculate the buoyancy term as a residual, but the authors choose not to do this which is interesting. Instead they calculate the time tendency by taking a 5-second finite time-difference on a signal that appear to have a 20-minute averaging window. So, I can't help wondering what would happen if that time difference was taken over one minute or maybe even 20 minutes. The way this was done means that the magnitude of the residual – which is assumed to be the buoyancy production – could have been substantially smaller if the time difference had been substantially larger. Maybe taking the time difference over another time window has a small effect; maybe one could even assume that the time tendency is negligible.

**Either way I would have preferred to: 1) Never say you measure something that is not actually measured, instead use "derive" or estimate"; 2) all terms in the budget be derived over the same time Reynolds averaging time window and; 3) describe carefully the way the turbulence terms are estimated in the Reynolds averaging for both sonic and lidar measurements.**

Moving on to the dissipation term, the whole theory behind the suggested way of estimating the dissipation rate relies on the Kolmogorov theory and the existence of an inertial sub-range in the power spectra. Here the authors deviate from this by allowing the exponent to deviate from the expected -5/3 slope; one of two criteria for determining that such a range to exist in their data. They compare the results to observations from the sonic anemometers with good results, but they never discuss how the latter was estimated. Presumably it was estimated using the same type of method since the sonic sampled at 10 Hz cannot actually measure the dissipation directly. However, with an 0.2 Hz sampling rate for the lidar, it seems more likely that deviations from the expected slope that is discussed is due to never actually reaching the inertial sub-range and not from actual deviations of the exponent as

explained in the text. Still the comparison to sonic observations, that sampled at 10 Hz, looks good, so why is that? If the estimates from the sonic are done for the same frequencies as for the lidar, maybe both estimates are wrong and that is why they compare so well.

To believe in this as a robust method, we should at least get to see some spectra from both sonic and lidar and time series of the calculated exponents (also from both) along with a discussion of what this would mean in case the lidar measurements at 0.2 Hz never reaches into the inertial subrange. If it does not reach into the inertial sub-range, the Kolmogorov no longer applies and Equation (2) cannot be used – at all. If it it just barely reaches this frequency range, equation 2 would allow calculation of dissipation from a single-frequency spectral estimate assuming the exponent is indeed prescribed to be the expected -5/3. I wonder what the results from such an approach would give?

**Hence for the dissipation rate estimates I would like: 1) To know if both instruments are interrogated using the same technique with the same parameters or, if not, what was the difference(s); 2) to see examples of power spectra from both instruments to be able to judge if the spectral estimates used are indeed inside the inertial subrange and; 3) see time series of the value of the exponent arising from the method the authors claim to have used.**

*Detailed comments*

Lines 17-19: Please repharse, as the buoyancy is in fact not measured by the lidar.

Lines 33-34: Langauge: "… terms, which are key physical quantities …"

Line 43: Maybe this is nit-picking but all these parameterization attempts to *model* turbulence. To simulate turbulence you need a DNS or possibly (stretching a bit) LES.

Line 66 and elsewhere: Why do you not start out with the full reference and then skip it at the end: "Nilsson et al. (2016a) used …".

Lines 84-85: Not to mention the cost of maintaining a tower compared to running a lidar!

Section 3.4: Why are we not shown a comparison between the shear-production terms instead of the shear stress terms?

Line 305: What do you mean by "gleaned"? Did you not actually measure this term?

Figures 9 & 11: There are some regions of these plots where the results seem quite noisy. Is there a way by which you can quality control these results, maybe looking at signal strength or pulse return rates?

Figures 10 & 12: By definition these profiles must add up to zero; that doesn't seem to always be the case, especially with in Figure 12a

---

## Author Response (AR2)

**Egusphere-2024-2163- Response Letter 3**

Dear Editor,

We would like to thank the editor for his comments that have allowed us to further clarify some aspects of the manuscript in this revised version. Hereafter, we report editor's comments and our replies (*in italics*). For yours convenience we have put the corresponding major changes introduced in red color in the revised version of the manuscript.

**Comments:**

In this paper the authors use a wind lidar to derive the terms in the TKE-budget. The two reviewers both had objections to some of the authors text and explanations and the authors responded to these in an adequate fashion. I have now reviewed the paper and I have some major concerns that I want to have answered before finally deciding on this paper.

**Major comments**

I am a big fan of using remote sensing instruments for this type of study and I think the time-height distributions shown in the last section of the paper illustrates their potential. There are, however, two potential major flaws in the representation of the TKE budget terms, especially relating to the dissipation term and the buoyancy term. Most of the comments below are based on the authors not properly explaining the choice of parameters in their Reynolds averaging for the two different instruments. I would also point out that with one of the terms in the budget equation, the dissipation, being estimated indirectly and another being calculated as a residual, the average budget profiles, like in Figure 10 & 12 must balance to zero; hence, if the dissipation is larger than the shear production, the buoyancy has to pick up the rest to balance the results. Therefore, Figure 12a is puzzling since that doesn't seem to happen here.

Starting with the buoyancy term, this is - contrary to the authors claims - never measured; it is estimated as a residual – after ignoring the pressure transport term which in my opinion is OK. But the reason to believe in the method comes down to the residual – assumed to be buoyancy – being much larger compared to other leading terms; that the residual cannot be explained in terms of errors in the other terms.

One could have assumed the time tendency term to be small and then proceed to calculate the buoyancy term as a residual, but the authors choose not to do this which is interesting. Instead they calculate the time tendency by taking a 5-second finite time-difference on a signal that appear to have a 20-minute averaging window. So, I can't help wondering what would happen if that time difference was taken over one minute or maybe even 20 minutes. The way this was done means that the magnitude of the residual – which is assumed to be the buoyancy production – could have been substantially smaller if the time difference had been substantially larger. Maybe taking the time difference over another time window has a small effect; maybe one could even assume that the time tendency is negligible.

Response: *As the reviewer suggests, we present the spatiotemporal distributions of turbulent kinetic energy, tendency term, turbulent transport term, dissipation rate, shear generation term, and buoyancy generation term under finite time differences of 5 seconds, 1 minute, 5 minutes, 20 minutes, and 25 minutes, as shown in Figures 11 to 15. It can be seen that as the difference time increases, the tendency term tends to stabilize, and even approaches 0 most of the time. When reaching 25 minutes, the results of the tendency term and buoyancy generation term become very different. Therefore, we can conclude that the smaller the difference time, the smaller the error between the tendency term and the buoyancy generation term.*

[Figure]

**Figure 11.** Temporal and spatial distributions of the TKE (a), tendency term (b), turbulent transport term (c), dissipation rate (d), shear generation term (e), and buoyancy generation term (f) on October 5, 2022 (a 5-second finite time-difference).

[Figure]

**Figure 12.** Temporal and spatial distributions of the TKE (a), tendency term (b), turbulent transport term (c), dissipation rate (d), shear generation term (e), and buoyancy generation term (f) on October 5, 2022 (a 1- minute finite time-difference).

[Figure]

**Figure 13.** Temporal and spatial distributions of the TKE (a), tendency term (b), turbulent transport term (c), dissipation rate (d), shear generation term (e), and buoyancy generation term (f) on October 5, 2022 (a 5- minute finite time-difference.

[Figure]

**Figure 14.** Temporal and spatial distributions of the TKE (a), tendency term (b), turbulent transport term (c), dissipation rate (d), shear generation term (e), and buoyancy generation term (f) on October 5, 2022 (a 20- minute finite time-difference).

[Figure]

**Figure 15.** Temporal and spatial distributions of the TKE (a), tendency term (b), turbulent transport term (c), dissipation rate (d), shear generation term (e), and buoyancy generation term (f) on October 5, 2022 (a 25- minute finite time-difference).

**Either way I would have preferred to: 1) Never say you measure something that is not actually measured, instead use "derive" or estimate"; 2) all terms in the budget be derived over the same time Reynolds averaging time window and; 3) describe carefully the way the turbulence terms are estimated in the Reynolds averaging for both sonic and lidar measurements.**

Response: (*1*): *As the reviewer suggests, we have made modifications in the revised version. (See lines 18 and 247) .*

(*2*) *and* (*3*): *In this study, for wind lidar, the time resolution of all budget terms is the same, which is about 20 min. We have added the text in the revised version*, "*These fluctuation components can be obtained by subtracting the average of the observed wind speed data within a time window of duration N. In the subsequent estimation of turbulent energy dissipation rate, the value of N needs to simultaneously meet the requirements of the Fast Fourier Transform (FFT) method. In discrete FFT, $2^m$ data points are required, where m is an integer. Therefore, in this paper, we conducted FFT calculations for $2^8 \times 5$ s points (i.e., approximately $N \approx 20$ min) using the data obtained from the wind lidar. All budget terms use the same duration window N. For the ultrasonic anemometer, $2^{14} \times 0.1$ s points (i.e., approximately 27 min) are required.*" (*See lines 191 -196*)

Moving on to the dissipation term, the whole theory behind the suggested way of estimating the dissipation rate relies on the Kolmogorov theory and the existence of an inertial sub-range in the power spectra. Here the authors deviate from this by allowing the exponent to deviate from the expected -5/3 slope; one of two criteria for determining that such a range to exist in their data. They compare the results to observations from the sonic anemometers with good results, but they never discuss how the latter was estimated. Presumably it was estimated using the same type of method since the sonic sampled at 10 Hz cannot actually measure the dissipation directly. However, with an 0.2 Hz sampling rate for the lidar, it seems more likely that deviations from the expected slope that is discussed is due to never actually reaching the inertial sub-range and not from actual deviations of the exponent as explained in the text. Still the comparison to sonic observations, that sampled at 10 Hz, looks good,

so why is that? If the estimates from the sonic are done for the same frequencies as for the lidar, maybe both estimates are wrong and that is why they compare so well.

To believe in this as a robust method, we should at least get to see some spectra from both sonic and lidar and time series of the calculated exponents (also from both) along with a discussion of what this would mean in case the lidar measurements at 0.2 Hz never reaches into the inertial subrange. If it does not reach into the inertial sub-range, the Kolmogorov no longer applies and Equation (2) cannot be used – at all. If it it just barely reaches this frequency range, equation 2 would allow calculation of dissipation from a single-frequency spectral estimate assuming the exponent is indeed prescribed to be the expected -5/3. I wonder what the results from such an approach would give?

**Hence for the dissipation rate estimates I would like: 1) To know if both instruments are interrogated using the same technique with the same parameters or, if not, what was the difference(s); 2) to see examples of power spectra from both instruments to be able to judge if the spectral estimates used are indeed inside the inertial subrange and; 3) see time series of the value of the exponent arising from the method the authors claim to have used.**

Response: *(**1**): For ultrasonic anemometers and wind lidar, the same technology is used in the calculation of each budget term. The only difference is the length of the time window, which is mainly due to the inconsistency in resolution between the two. As the reviewer suggests, we have made modifications in the revised version. "These fluctuation components can be obtained by subtracting the average of the observed wind speed data within a time window of duration N. In the subsequent estimation of turbulent energy dissipation rate, the value of N needs to simultaneously meet the requirements of the Fast Fourier Transform (FFT) method. In discrete FFT, $2^m$ data points are required, where m is an integer. Therefore, in this paper, we conducted FFT calculations for $2^8 \times 5$ s points (i.e., approximately $N \approx 20$ min) using the data obtained from the wind lidar. All budget terms use the same duration window N. For the ultrasonic anemometer, $2^{14} \times 0.1$ s points (i.e., approximately 27 min) are required." (See lines 191 -196)*

*(**2**) **and** (**3**): Thanks for the editor's professional comments. In our previous researches, we demonstrated that in most weather conditions (e.g. sunny or cloudy), turbulence spectrum estimated by wind lidar can reach the inertial subrang. In some cases, the turbulence spectrum estimated by wind lidar cannot reach the inertial subrang, resulting in statistical errors (Xian et al., 2024c). This is mainly due to the detection frequency of wind lidar being only 0.2 Hz. This also provides a basis for increasing the detection frequency of wind lidar in the future. Here, we present our previous research findings as shown in Figure 2. It provides a comparison of turbulence spectra in three directions (U, V, W) between wind lidar and ultrasonic anemometer. From the graph, we can see that there is a high degree of consistency between the two in all three directions, and they all conform well to the -5/3 pow law. This proves that wind lidar can effectively monitor the wind speed turbulence spectrum in the inertial subrang. Furthermore, we present a comparison of the power exponents estimated by wind lidar and ultrasonic anemometer, as shown in Figure 3. From figures 3(d)- (f), it can be seen that both are quite consistent with the -5/3 power law. We cited two references about our previous research work in the paper. We have also added the texts in the revised version. (See lines 225 -230)*

*For more information, please refer to the papers:*

*Xian, J. H., Lu, C., Lin, X. L., Yang, H. L., Zhang, N., and Zhang, L.: Directly measuring the power-law exponent and kinetic energy of atmospheric turbulence using coherent Doppler wind lidar, Atmospheric Measurement Techniques, 17, 1837-1850, 2024.*

*Xian, J., Luo, H., Lu, C., Lin, X., Yang, H., and Zhang, N.: Characteristics of the atmospheric boundary layer height: A perspective on turbulent motion, Science of The Total Environment, 919, 170895, 2024.*

[Figure]

**Figure 2.** Comparison of the turbulence spectra obtained with the wind lidar and the ultrasonic anemometer in three directions: (a) *U*, (b) *V*, and (c) *W*, and the corresponding correlations (d), (e), and (f).

[Figure]

**Figure 3.** Comparison of the turbulent kinetic energy obtained from the wind lidar and three-dimensional ultrasonic anemometer on **January 1, 2022** in the (a) *U*, (b) *V*, and (c) *W* directions and the power-law exponent distribution in the (d) *U*, (e) *V*, and (f) *W* directions.

**Detailed comments**

Lines 17-19: Please rephrase, as the buoyancy is in fact not measured by the lidar.

Response: *As the reviewer suggests, we have made modifications in the revised version. (See lines 18 and 253).*

Lines 33-34: Langauge: "… terms, which are key physical quantities …"

Response: *As the reviewer suggests, we have made modification in the revised version. (See line 34).*

Line 43: Maybe this is nit-picking but all these parameterization attempts to model turbulence. To simulate turbulence you need a DNS or possibly (stretching a bit) LES.

Response: *Thanks for the editor's professional comments. We have added the texts in the revised version. (See lines 38-40).*

Line 66 and elsewhere: Why do you not start out with the full reference and then skip it at the end: "Nilsson et al. (2016a) used …".

Response: *As the reviewer suggests, we have revised the citation way of this reference. (See line 69).*

Lines 84-85: Not to mention the cost of maintaining a tower compared to running a lidar!

Response: *As the reviewer suggests, we have added the texts in the revised version. (See line 86).*

Section 3.4: Why are we not shown a comparison between the shear-production terms instead of the shear stress terms?

Response: *Due to Δz = 160 m, the shear generation term obtained by the ultrasonic anemometer has a relatively large error. Therefore, we compared the u'w' and v'w' obtained by the wind lidar and ultrasonic anemometer. As the reviewer suggests, we have added the texts in the revised version. (See lines 250-252).*

Line 305: What do you mean by "gleaned"? Did you not actually measure this term?

Response: *As the reviewer suggests, we have modified the text in the revised version. (See line 314).*

Figures 9 & 11: There are some regions of these plots where the results seem quite noisy. Is there a way by which you can quality control these results, maybe looking at signal strength or pulse return rates?

Response: *Thanks for the editor's professional comments. Currently, in data processing, the signal-to-noise ratio has been used to remove noise points. We will further optimize the noise removal method in future research.*

Figures 10 & 12: By definition these profiles must add up to zero; that doesn't seem to always be the case, especially with in Figure 12a.

Response: *As the reviewer suggests, we have modified Figures 10, 12, and 14 and added the indicator line of Et-B-S-T+D in the figures, which can clearly and intuitively show the balance of turbulent kinetic energy.*

On behalf of all authors,
Sincerely,
Honglong Yang

Shenzhen National Climate Observatory
Meteorological Bureau of Shenzhen Municipality
518000 Shenzhen, China
E-mail:  yanghl01@163.com